# A strawberry accession with elevated methyl anthranilate fruit concentration is naturally resistant to the pest fly *Drosophila suzukii*

**Lasse B. Bräcker**[1], **Xiaoyun Gong**[2], **Christian Schmid**[3], **Corinna Dawid**[3], **Detlef Ulrich**[4], **Tuyen Phung**[1], **Alexandra Leonhard**[1], **Julia Ainsworth**[1], **Klaus Olbricht**[5,6]*, **Martin Parniske**[2]*, **Nicolas Gompel**[1]*

**1** Chair of Evolutionry Ecology, Faculty of Biology, Ludwig-Maximilians Universität München, Planegg-Martinsried, Germany, **2** Chair of Genetics, Faculty of Biology, Ludwig-Maximilians Universität München, Planegg-Martinsried, Germany, **3** Chair of Food Chemistry and Molecular Sensory Science, Technical University of Munich, Freising, Germany, **4** Julius Kühn-Institute, Federal Research Centre for Cultivated Plants, Quedlinburg, Germany, **5** Hansabred GmbH & Co. KG, Dresden, Germany, **6** Humboldt-Universität zu Berlin, Faculty of Life Sciences, Albrecht Daniel Thaer-Institute of Agricultural and Horticultural Sciences, Berlin, Germany

* k.olbricht@hansabred.org (KO); parniske@lmu.de (MP); gompel@bio.lmu.de (NG)

**Data Availability Statement:** All relevant data are within the paper and its Supporting Information files.

## Abstract

During the past decade, *Drosophila suzukii* has established itself as a global invasive fruit pest, enabled by its ability to lay eggs into fresh, ripening fruit. In a previous study, we investigated the impact of different strawberry accessions on the development of *D. suzukii* eggs, in the search of natural resistance. We identified several accessions that significantly reduced adult fly emergence from infested fruit. In the present study, we aimed at understanding the chemical basis of this effect. We first noted that one of the more resistant accessions showed an unusual enrichment of methyl anthranilate within its fruit, prompting us to investigate this fruit compound as a possible cause limiting fly development. We found that methyl anthranilate alone triggers embryo lethality in a concentration-dependent manner, unlike another comparable organic fruit compound. We also showed that a chemical fraction of the resistant strawberry accession that contains methyl anthranilate carries some activity toward the egg hatching rate. Surprisingly, in spite of the lethal effect of this compound to their eggs, adult females are not only attracted to methyl anthranilate at certain concentrations, but they also display a concentration-dependent preference to lay on substrates enriched in methyl anthranilate. This study demonstrates that methyl anthranilate is a potent agonist molecule against *D. suzukii* egg development. Its elevated concentration in a specific strawberry accession proven to reduce the fly development may explain, at least in part the fruit resistance. It further illustrates how a single, natural compound, non-toxic to humans could be exploited for biological control of a pest species.

**Funding:** The present work has been financed with funding from the Ludwig-Maximilians University of Munich (NG and MP). KO is an employee of Hansabred GmbH. The funder provided support in the form of salaries for author KO, but did not have any additional role in the study design, data collection and analysis, decision to publish, or preparation of the manuscript. The specific roles of this author are articulated in the author contributions' section.

**Competing interests:** KO is an employee of Hansabred GmbH. The affiliation of KO to Hansabred GmbH does not alter our adherence to PLOS ONE policies on sharing data and materials.

## Introduction

The agricultural production of several fruits and berries of economical importance has recently become threatened by a pest fly, *Drosophila suzukii*. The economical incidence of this pest amounts to hundreds of millions of euros or dollars, worldwide. The costs entail loss of harvest as well as pest management strategies. In spite of some progress, pest management strategies remain insufficient to contain the fly [1]. The management strategies are diverse and explore or exploit the use of gene drive [2], parasitoids and predators [3, 4], insecticides, mulching [5], or natural variation in egg-laying substrate suitability [6, 7]. In a previous study [7], we investigated the existence of natural resistance to *Drosophila suzukii* among species and cultivars of strawberries (*Fragaria sp.*).

*Drosophila suzukii*, a close relative to the model species *D. melanogaster*, has spread from East Asia to the Western World during the years 2000 [8]. It abounds during the spring, the summer or the fall in agricultural regions producing berries. Upon mating, *D. suzukii* females search for their preferred egg-laying substrates, ripening or ripe fruits [9, 10], pierce the fruit surface with their elongated ovipositor [11, 12], and push an egg into the fruit. This action has two damaging consequences for the fruit. The ovipositor is a vector for all superficially attached microorganisms and the piercing moves these deep into the fruit thus accelerating decay. Second it will soon be infested with a larva feeding on it.

We reasoned that strawberries with different genetic makeup may produce fruits of variable suitability for the development of a fly's offspring. In our original study [7], we evaluated the fly emergence rate from fruits of 111 *Fragaria* accessions, after exposing their ripe fruits to fertilized females for a fixed time. We found an impressive variation in emergence rate, from highly susceptible accessions with tens of flies emerging from single fruits, to nearly resistant accessions, from which hardly any fly progeny emerged although the females deposited eggs in them [7]. While this variation correlated mainly with the size of the berries, we identified accessions that departed from this trend, suggesting that other factors may influence the quality of the fly breeding substrate. In the present study we focused on one of these notably resistant accessions, 300 (referred to as Großolbersdorf in [13]), to investigate the origin of the low emergence rate. The emergence rate from 300 berries was significantly lower than what the size of its berries predicted [7]. To our surprise, the low emergence rate did not result from a reluctance of females to lay on these fruits, but rather from apparent toxicity of the fruits, limiting embryo hatching as well as larval survival. We have identified a chemical compound present at elevated concentration in the resistant accession with a dual role in the interaction with *D. suzukii*; it was attractive to the female flies in the context of egg laying and at the same time may kill the fly embryos in the infested fruits.

## Materials and methods

### Analysis of strawberry volatiles: Extraction of fruit volatiles by immersion stir bar sorptive extraction (imm-SBSE)

To prepare an enzyme-inhibited strawberry juice, ripe frozen fruits of each accession or cultivar of the whole season were pooled. One mass part of fruits without sepals was homogenized in one volume part of a solution of 18.6% (m/v) NaCl in distilled water with a household blender for 2 minutes. The homogenate was centrifuged at 4000 rpm for 30 minutes. One 100 ml of the supernatant were mixed with 10 μl internal standard solution (0.1% (v/v) 2,6-dimethyl-5-hepten-2-ol dissolved in ethanol). For each sample, three head-space vials containing 3 g natrium chloride each for saturation were filled with 10 ml of supernatant, sealed with magnetic crimp caps including septum, and stored at 4°C for up to three weeks.

An aliquot of 8 ml of the saturated homogenate without the solid natrium chloride deposit was transferred in an empty glass vial for volatile isolation by SBSE. A stir bar with 0.5 mm film thickness and 10 mm length coated with polydimethysiloxan (PDMS) was placed in the liquid (Gerstel, Mülheim an der Ruhr, Germany). The stir bar was moved at 350 rpm at room temperature for 45 min. After removal from the strawberry juice, the stir bar was rinsed with purified water, gently dried with a lint-free tissue and then transferred into a glass tube for thermal desorption and subsequent gas chromatography (GC) analysis.

## Analysis of strawberry volatiles: Gas chromatography, mass spectrometry

The following parameters were used for the thermal desorption unit (TDU, Gerstel) and the cold injection system (CIS4, Gerstel): thermal desorption at 250˚C, cryo-trapping at -150˚C. The TDU-CIS4 unit was used in Gerstel-modus 3: TDU splitless and CIS4 with 15 ml/min split flow. The analyses were performed with an Agilent Technologies 6890N gas chromatograph equipped with an Agilent 5975B quadrupol MS detector. Compounds were separated on a polar column ZB-Wax plus 30 m length x 0.25 mm ID x 0.5 μm film thickness. Helium was used as a carrier gas with a column flow rate of 1.1 ml/min. Temperature programme: 45˚C (3 min), temperature gradient 3 K/min to 210˚C (30 min). The mass spectrometer was used with electron ionization at 70 keV in full scan mode. All samples were run with two analytical repetitions from an identical part of the supernatant.

## Analysis of strawberry volatiles: Data processing

The resulting total ion chromatograms (TIC) were integrated using the Agilent ChemStation™ routine with an initial threshold of 15 counts for all samples. The value of the initial threshold was chosen in such a manner that the most intense chromatograms contain about 100 integrated peaks. The integration results were imported subsequently as raw data (txt-formatted) into the chemometrical software ChromStatTM 2.6 from Analyt-MTC (Müllheim, Germany). The semi-quantitative results were expressed as non-dimensional values (counts or relative concentrations).

## Fly rearing

For all experiments, we used our *D. suzukii* wild type stock "Lyon", which is derived from an isofemale line collected in Lyon, France by Roland Allemand. Flies were kept on standard corn flour-based fly food at 23˚C, with a relative humidity (rH) of 50% in a 12:12 hour light and dark cycle. For all behavioral experiments, 0–1 day-old flies were collected daily and aged for 5–6 days before the experiment.

## Embryo survival assay

To collect eggs of *D. suzukii*, mixed groups of at least a hundred 4-day-old flies were transferred to egg-laying cages. The bottom of these cages was formed by exchangeable Petri dishes filled with 1% agarose and a 2 mm thick layer of a saturated starch solution made with commercially available mango juice. These plates were prepared fresh before egg collection and allowed to dry until the starch solution had hardened but no longer sticky. Every 20 minutes, the plates were exchanged and the eggs were washed out of the starch layer by rinsing it with water into a collection mesh.

To create an environment for embryonic development, 1.5 ml of 1% agar was poured into 12- well cell culture plates (VWR). After the medium temperature cooled to below 50˚C, pre-diluted methyl anthranilate (MA) or isoamyl acetate in paraffin oil was pipetted and mixed

into each well, reaching the indicated final concentrations of MA (Fig 2; stick-and-balls models of MA and isoamyl acetate were generated from http://molview.org).

For the test of strawberry fraction toxicity, each fraction was resolubilized in water using a shaker. We aimed at increasing the concentration of metabolites by a factor of ten in the medium onto which the embryos were tested, compared to their native concentration in the fruit (see Results). We calculated this dilution, on the measurement of an average 80% moisture content in *Fragaria vesca* fruits. We therefore prepared a 20x concentrated solution of each fraction that we combined with a 2% agar solution to reach a 10x fraction concentration in 1% agar medium. 0.5 ml of this medium was poured in the wells of 24-well cell culture plates (VWR).

Once the medium was fully solidified, 8 eggs were transferred into each well using a brush. Wells were then covered with moist tissue paper, and the plates were transferred into climate chambers at 23°C and 50% rH. Egg hatching was counted at the indicated intervals. In total, 36 wells were counted for each MA concentration, and 24 wells for each isoamyl acetate concentration.

Significance was determined using a non-parametric Kruskal wallis test and a Dunn's multiple comparisons test.

## Strawberry fractions

**Quantification of methyl anthranilate (1) by stable isotope dilution analysis (SIDA-UHPLC-MS/MS).** Methyl anthranilate concentrations in each fraction are summarized in Table 1. Methyl anthranilate (**1**) was analysed by means of a newly developed SIDA-UHPLC-MS/MS$_{MRM}$ method. Therefore, the deuterium-labelled methyl anthranilate-d$_3$ (1-d$_3$) was utilized.

**Internal standard (IS).** A stock solution (500 μL) of methyl anthranilate-d$_3$ (**1-d₃**) (2.53 mmol/L) was prepared in CDCl$_3$, its exact concentration was verified by means of quantitative NMR (qNMR) and it was stored at −20°C until used.

**Sample preparation.** Strawberries were frozen in liquid nitrogen and pulverized by a knife mill (GM200, Retsch GmbH, Haan, Germany). Crushed samples were weighed and extracted three times with variable amounts of ethyl acetate, methanol/water (70/30, v/v) and water by homogenizing with an Ultra Turrax T25 digital (IKA GmbH & Co. KG, Staufen, Germany). After each extraction step, samples were filtered, freed from organic solvents and freeze-dried. Sample weights, solvent amounts and extraction yields are stated in S1 Table. Lyophilized extracts (1–13 mg) were solved in 1 mL EtOAc (EtOAc extract) or in 1 mL MeOH/H$_2$O (50/50, v/v) (MeOH/H$_2$O and H$_2$O extract) in addition with internal standard (methyl anthranilate-d$_3$, 1-d$_3$) to a final concentration of 0.253 μmol/L.

**Ultra high performance liquid chromatography-mass spectrometry (UHPLC-MS/MS).** A QTRAP 6500 mass spectrometer (Sciex, Darmstadt, Germany) was used and operated in the multiple reaction monitoring (MRM) mode (ion spray voltage, 5500 V): curtain gas, 35 V; temperature, 500°C; gas 1, 55 V; gas 2, 65 V; collision-activated dissociation gas, medium; and entrance potential, −10 V. For compound optimization flow injection with a syringe pump (10 μL/min) and compound solutions in ACN (0.1% FA) were used. The samples were separated by means of a Nexera X2 UHPLC (Shimadzu Europa GmbH, Duisburg, Germany) consisting of two LC pump systems LC-30AD, a DGU-20A5 degasser, a SIL-30AC autosampler, a CTO-30A column oven, and a CBM-20A controller and equipped with a Kinetex 1.7 μm C18 column (100 × 2.1 mm, 100 Å, Phenomenex). Chromatography was performed using an injection volume of 1 μL, a flow rate of 0.5 mL/min and a column temperature of 55°C. The solvent system consisted of A: formic acid (0.1% in water, pH 3.5) and B: acetonitrile (0.1% formic acid). The following gradient was used: 0 min, 5% B; 0.5 min, 5% B; 7 min, 70% B; 7.25 min,

**Table 1. Methyl anthranilate concentration in strawberry fractions.**

*methyl anthranilate µg/g extract*

| fraction/sample | concentration | | |
|---|---|---|---|
| EtOAc #2 | 1072.69 | +/- | 8.67 |
| EtOAc #4 | 890.26 | +/- | 9.14 |
| EtOAc #5 | 119.48 | +/- | 0.49 |
| EtOAc #6 | 369.21 | +/- | 10.19 |
| MeOH/$H_2O$ #2 | 6.72 | +/- | 1.06 |
| MeOH/$H_2O$ #4 | 5.47 | +/- | 0.9 |
| MeOH/$H_2O$ #5 | 0.86 | +/- | 0.07 |
| MeOH/$H_2O$ #6 | 1.28 | +/- | 0.25 |
| MeOH/$H_2O$ #13 | 0.95 | +/- | 0.064 |
| $H_2O$ #2 | 21.82 | +/- | 0.53 |
| $H_2O$ #4 | 8.97 | +/- | 0.22 |
| $H_2O$ #5 | 2.08 | +/- | 0.007 |
| $H_2O$ #6 | 3.59 | +/- | 0.045 |
| $H_2O$ #13 | 1.79 | +/- | 0.013 |

*methyl anthranilate ng/g strawberries*

| fraction number | concentration | | |
|---|---|---|---|
| EtOAc #2 | 17301.52 | +/- | 139.91 |
| EtOAc #4 | 21674.02 | +/- | 222.45 |
| EtOAc #5 | 1175.73 | +/- | 4.83 |
| EtOAc #6 | 2847.38 | +/- | 78.60 |
| MeOH/$H_2O$ #2 | 764.66 | +/- | 120.77 |
| MeOH/$H_2O$ #4 | 446.38 | +/- | 73.29 |
| MeOH/$H_2O$ #5 | 56.87 | +/- | 4.63 |
| MeOH/$H_2O$ #6 | 90.28 | +/- | 17.37 |
| MeOH/$H_2O$ #13 | 252.07 | +/- | 17.04 |
| $H_2O$ #2 | 263.94 | +/- | 6.40 |
| $H_2O$ #4 | 65.55 | +/- | 1.60 |
| $H_2O$ #5 | 15.34 | +/- | 0.06 |
| $H_2O$ #6 | 23.07 | +/- | 0.29 |
| $H_2O$ #13 | 18.91 | +/- | 0.14 |

100% B; 8.25 min, 100% B; 8.5 min, 5% A; 10 min, 5% B. Data acquisition and instrumental control was performed with Analyst 1.6.3 software (Sciex, Darmstadt, Germany). After optimizing instrument settings, analytes and the internal standard were analyzed using the MRM transition Q1/Q3 of *m/z* 151.9/120.0 as quantifier (DP = 6 V, CE = 13 V, CXP = 14 V) and Q1/Q3 of *m/z* 151.9/92.0 as qualifier (DP = 6 V, CE = 31 V, CXP = 8 V) for methyl anthranilate (**1**) and Q1/Q3 of m/z 154.9/119.9 as quantifier (DP = 1 V, CE = 13 V, CXP = 16 V) and Q1/Q3 of *m/z* 154.9/92.1 as qualifier (DP = 1 V, CE = 31 V, CXP = 10 V) for methyl anthranilate-$d_3$ (**1-$d_3$**).

**Calibration curve and linear range.** A stock solution of methyl anthranilate-$d_3$ (IS, 1-$d_3$, 2.53 mmol/L) and the analyte methyl anthranilate (1, 14.92 mmol/L) was prepared in $CDCl_3$, and its exact concentration was verified by means of quantitative NMR (qNMR). Thereafter, different analyte concentrations (1, 0.149; 0.298; 1.492; 2.984; 14.92; 29.84 µmol/L) were mixed with constant concentrations of the IS (1-$d_2$, 0.253 µmol/L). Triplicate UHPLC-MS/MS analysis calibration curve was prepared by plotting peak area ratios of methyl anthranilate (1) to the internal standard methyl anthranilate-$d_3$ (1-$d_3$) against concentration ratios of 1 to the IS using linear regression. The response was linear with a correlation coefficient of >0.99 for

chosen molar ratios and the contents of methyl anthranilate (1) in the samples were calculated using the respective calibration function. Determination of the limit of detection (LOD) at a signal-to-noise ratio of 3 and the limit of quantitation (LOQ) at a signal-to-noise ratio of 10 revealed the following values: LOD: ≤0.002 μM; LOQ ≤0.01 μM.

## Strawberry fractions

**Nuclear magnetic resonance spectroscopy (NMR).** One-dimensional $^1$H quantitative NMR (qNMR) experiments were acquired on a 400 MHz Avance III spectrometer equipped with a Double Resonance Broadband probe (Bruker, Rheinstetten, Germany) as reported by Frank et al. [14]. Chemical shifts are reported in parts per million, relative to solvent signal of CDCl$_3$ (7.26 ppm). All pulse sequences were taken from Bruker software library. For data processing Topspin NMR software (version 3.2; Bruker, Rheinstetten, Germany) was used.

**Measuring moisture content of strawberry fruits by LOD (loss on drying).** Ripe fruits of several *F. vesca* were harvested as in [7] and dried at 60˚C. The mass of each fruit was measured before and at different time points during the drying process. Loss of mass was saturated after 48 h.

## Egg-laying preference assay

To measure egg-laying preference, Petri dishes with two different egg-laying substrates were prepared by blending commercially available strawberries (purchased at a local supermarket) and cooking the resulting purée with 1% agarose. The hot mixture was poured into the Petri dishes and left to harden; one half was then removed and the empty half was filled with the same purée and agarose mixture, supplemented with MA. In total, 20 ml of substrate was used per plate. MA was only added to the purée-agarose mixture after it had cooled to below 50˚C.

Mixed groups of 5–6 day-old males and females were transferred onto the Petri dishes using egg-laying cages. After 16 hours of egg-laying at 23˚C and 50% rH, flies were removed and the eggs on each half were counted. Results were used to calculate the oviposition preference index defined by (#eggs on control side–#eggs on MA side)/(#eggs on control side + #eggs on MA side). In total, egg laying on 12 individual plates was evaluated for each concentration. Significance was determined by one-sample t-tests against a hypothetical mean of 0.

## Trap assay

5–6 day-old groups of a hundred male and female flies were transferred to 19x32x10 cm plastic boxes covered with a lid and fitted with a mesh for aeration. Two traps were placed in each cage. They were made by covering 20 ml polyethylene flasks (5.5 cm high x 2.5 cm wide) with Whatman filter paper, sealing it with tape, and then piercing the paper with a conical 1000 μl pipette tip, so that the broad end of the tip abutted the paper and the distal end of the tip extended into the trap. To allow flies to enter the trap, the distal end of the tip was cut to create a 2 mm diameter opening. Traps contained water or a water/MA mixture at the concentration indicated in Fig 4, as bait. After 18 hours, males and females in each trap were counted. Results were used to calculate the preference index defined by (#flies in control trap–#flies in MA trap)/(#flies in control trap + #flies in MA trap). In total, nine replicates per concentration were evaluated. Significance was determined by one-sample t-tests against a hypothetical mean of 0.

# Results

## High levels of methyl anthranilate in a resistant strawberry accession

We hypothesized that the resistance of strawberry accessions to *D. suzukii* may be correlated with the chemical composition of their fruits. To identify potential chemical signatures of the

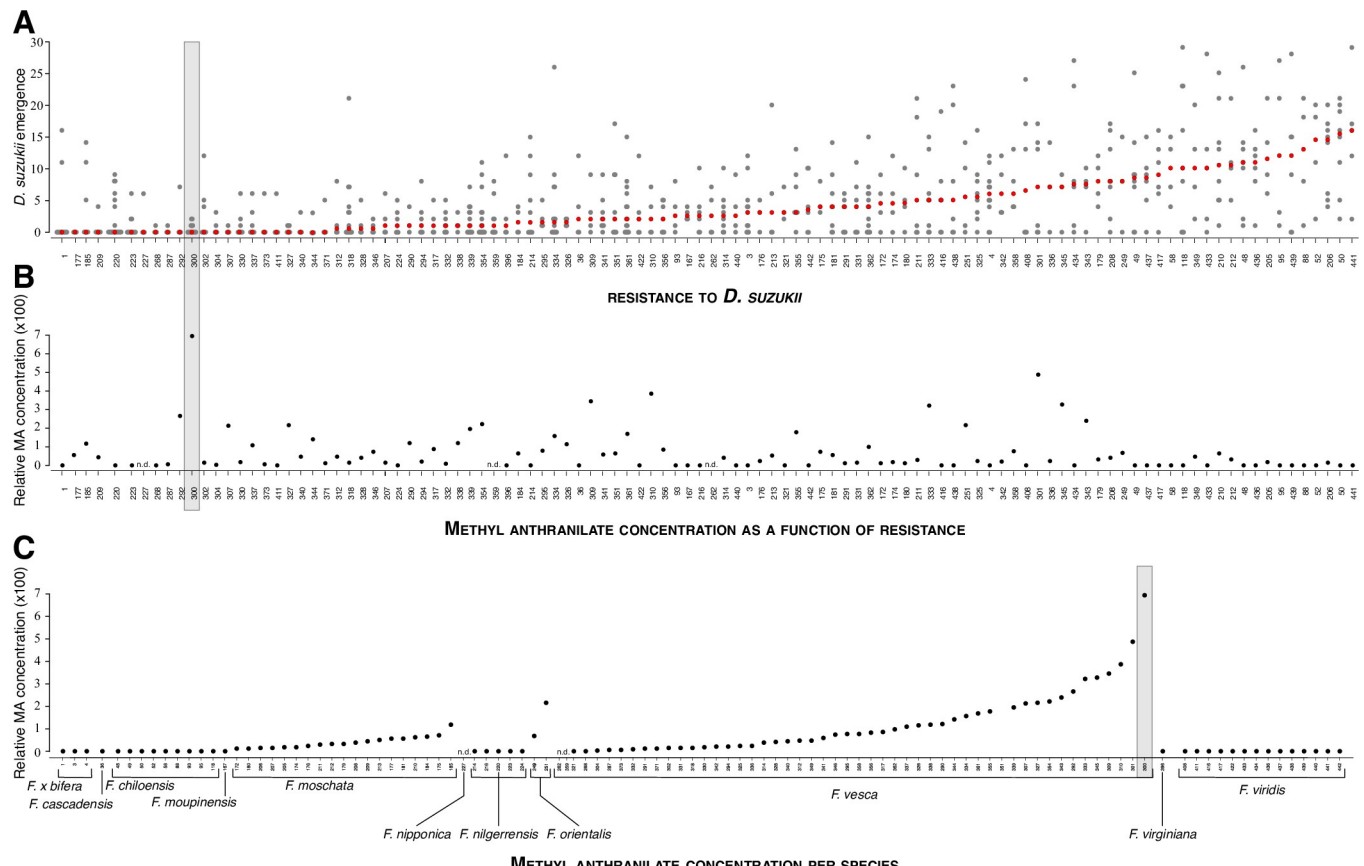

**Fig 1. Methyl anthranilate concentration in strawberry accessions with various degrees of resistance to *Drosophila suzukii*.** (A) *Fragaria* accessions differ in their resistance to *Drosophila suzukii* (replotted from [7]). (B) MA concentration is variable across *Fragaria* accessions. The *Fragaria* accession with highest MA levels, 300, is also one of the most resistant to *D. suzukii* (gray shading). (C) MA concentration is variable in some *Fragaria* species.

resistance, we determined the relative concentrations of secondary compounds in mature fruits of accessions in the light of their susceptibility to *D. suzukii* (Fig 1A, redrawn from [7]). We noticed that fruits of a particularly resistant accession, 300, also had an exceptionally high concentration of a particular compound, methyl anthranilate (MA) (Fig 1C; S1 Table). MA is one of the key compounds in the aroma of woodland strawberry (*Fragaria vesca*), giving fruity, sweet, soapy or grape notes [15, 16]. Although we did not observe any general relationship between resistance to *D. suzukii* and MA levels (Fig 1B), the exceptionally high MA levels of 300 prompted us to examine a possible role of this molecule to the resistance of this very accession. Beyond this particularity of 300, and although other accessions show elevated MA, it is in fact not possible to run a correlation analysis, as MA concentration is calculated using a batch of fruits for each accession, but emergence is quantified for individual fruits. It is also worth noting that distinct mechanisms could underlie resistance in different accessions.

## Methyl anthranilate is toxic to *D. suzukii* embryos

To directly test a possible involvement of MA into the resistance, we monitored the development of *D. suzukii* eggs on a substrate containing MA. We transferred freshly laid *D. suzukii* eggs onto an agar-based medium, in the wells of a 12-well cell culture plate (8 eggs/well),

containing different amounts of MA. To measure embryo survival, we counted the number of hatched embryos in each well every day. Under suitable conditions, we expect the vast majority of wild-type *D. suzukii* eggs to have hatched after 48 hours. Our data confirmed this, with a 90% hatching rate in wells devoid of MA (Fig 2A). The effect of MA was significant at a concentration of $2.10^{-4}$ (v/v), where 50% of the embryos failed to hatch after 72 hours. The toxic effect reached saturation at a concentration of $5.10^{-4}$. Indeed, those embryos that survived had all hatched after 48 h, similar to control groups, with no additional embryos hatching after 72 h. We concluded that the reduced number of hatched embryos at 48 hours did not result from delayed embryonic development and that MA is toxic to *D. suzukii* eggs.

Given the fragility of embryonic development, the observed lethality may simply result from the high concentration of an organic compound in the environment. To probe the specificity of MA, we repeated the assay with another secondary compound. We chose the ester isoamyl acetate, a signature odor of banana [17], also found in a variety of different fruits, including strawberries [18]. Increasing concentrations of isoamyl acetate, however, had no effect on the embryo hatching rate (Fig 2B). Up to concentrations as high as $10^{-3}$ percent isoamyl acetate, we observed no difference in hatching. We concluded that the mere presence of an organic compound, even highly concentrated, was not necessarily toxic to embryos, and inferred that the effect of MA is therefore specific.

### Narrowing down the active principle of strawberry resistance

The high levels of MA in the resistant accession 300 and its toxicity to fly embryos in an artificial substrate raised the possibility that MA participates to the fruit resistance. Therefore, to narrow down the origin of 300 resistance, we proceeded to fractionate its fruits (Fig 3) and to test the toxicity of the fractions on embryos. While we grew 300 plants in our green house, the small population size yielded only low amounts of fruit material, limiting our experiments. For a broad separation, we first chose to fractionate the fruits with three solvents. We generated a water-soluble fraction ($H_2O$), a methanol/water fraction ($MetOH/H_2O$) and an ethyl acetate fraction (EtOAc). We first measured the MA concentration in fractions of three independent harvests and assessed variability between batches (Table 1; S1 Table). The highest levels of MA were detected in ethyl acetate fractions (1–20 µg/g strawberries), while the MA levels in methanol/water and water fractions were at least an order of magnitude lower (<0.1 µg/g strawberries). We used the fractions of an additional batch of fruits to test their activity on *D. suzukii* embryos in cell culture plates, as described above. If MA is the active compound within the fruit (or one of them), we expect to detect an effect on embryo survival that correlates with the amount of MA found in each fraction. Preparing medium for our assays in which compounds are at their native fruit concentration proved to be challenging. To mitigate a potential loss of activity due to loss or alteration of certain compounds during fractionation, and to account for variation between fruit batches, we decided to concentrate each fraction ten times relative to their native concentrations in the final medium. Reasoning that *Fragaria vesca* fruits contain about 80% water (see methods), we reconstituted each fraction with 8% water, together with agar and the other elements of the medium (see methods). We observed a 10% reduction in embryo hatching with the methanol/water fraction as well as with the ethyl acetate fraction, but no reduction, compared to control when exposing the eggs to the water fraction. By contrast, similar fractions from commercial (non-resistant) *Fragaria ananassae* showed no toxicity at all in the same assay (not shown). The effect of the 300 ethyl acetate fraction, although modest, compared to the effect measured previously in whole fruits, or the effect of pure MA, could, however, be due to MA itself. If this is correct, the similar effect of the methanol/water fraction must result from another compound, as MA levels in this fraction are comparably low

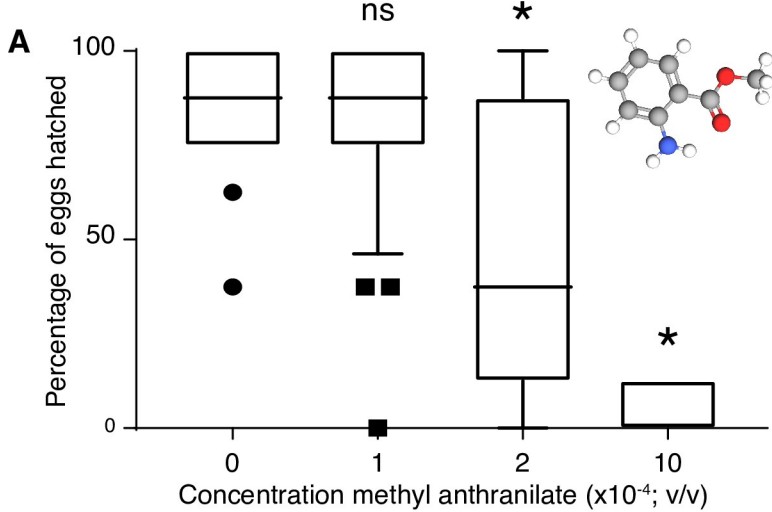

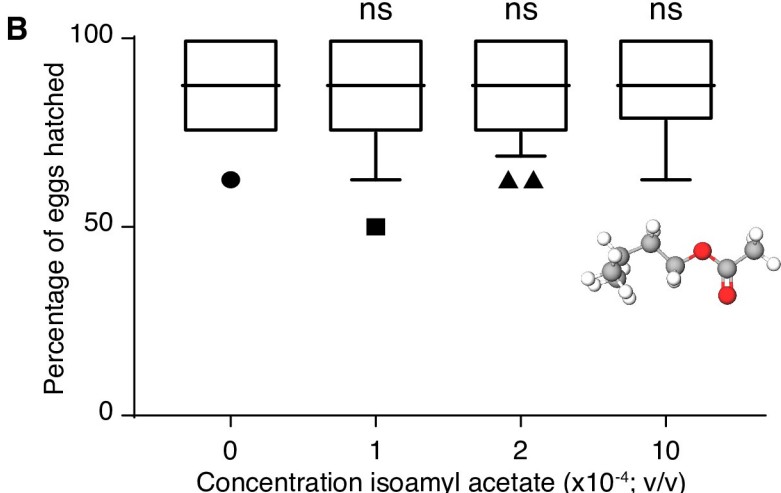

**Fig 2. *Drosophila suzukii* embryonic survival on methyl anthranilate and isoamyl acetate.** The box plots indicate the median number of embryos per well that have hatched on agar plates with various concentrations of MA (A) or isoamyl acetate (B) after 48 h. An MA concentration of $2.10^{-4}$ (v/v) or higher significantly reduces embryo survival as compared to the control group. No effect was observed when adding isoamyl acetate to the medium. (Box plots with median and 10–90 percentiles, n = 36 for MA, n = 24 for IA, stars indicate significant differences when compared to the control group; *** P-value<0.005, **** P-Value<0.0005 ns: non-significant, see methods for statistics).

to the harmless water fraction (Table 1; S1 Table). These results support the notion that MA may contribute to the embryonic lethality observed in 300 fruits.

## *D. suzukii* females preferences to lay on substrates containing methyl anthranilate vary with concentration

With MA having a strong effect on egg hatching, we next wondered how this molecule is perceived by adult female flies. We first examined its effect on females in the context of egg- laying. Given its toxicity to the eggs, we anticipated that females might avoid laying eggs into a substrate with increased MA levels. To test this hypothesis, we offered mixed groups of flies within a cage two different egg-laying substrates. These substrates consisted of strawberry

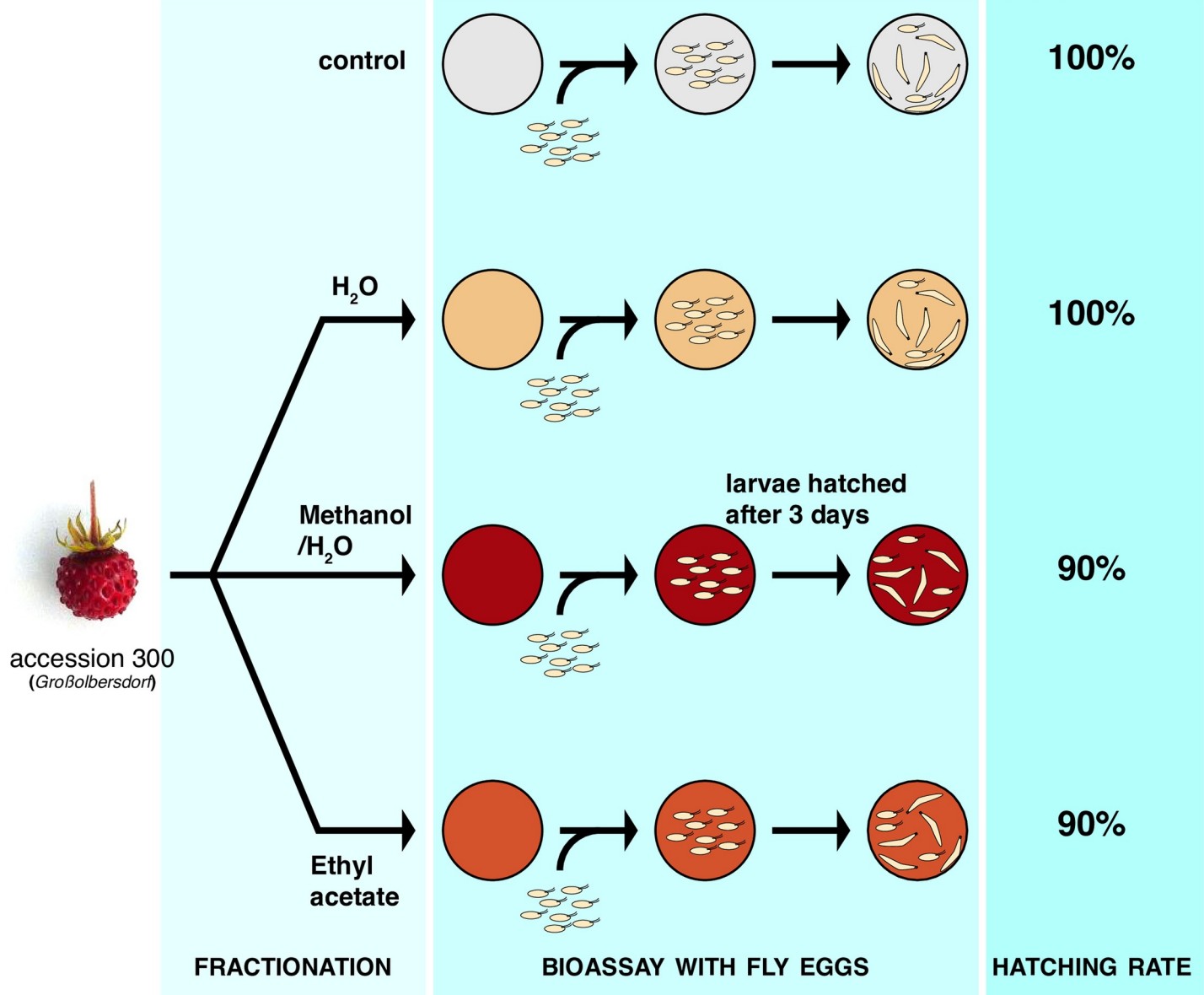

**Fig 3. Fractions of *Fragaria vesca* accession 300 carry antagonist egg activity.** The schematic depicts the experimental flow from fractionation to the assessment of fraction activity. Upon fractionation with three different solvents, the fractions were tested in agarose plates onto which fly embryos were let to develop. The hatching rate was reduced for methanol/water and ethyl acetate fractions, but not for the water fraction. All percentages were normalized to the hatching rate of the control.

purée [10] that was solidified with agarose and poured into two halves of a Petri dish (Fig 4, schematics). One half was filled with purée alone, while the other half was filled with the same purée supplemented with MA. We tested two MA concentrations, which resulted in contrasted egg-laying preferences (Fig 4). Females showed a clear aversion to lay on a substrate with a concentration of MA that would kill all their eggs ($5.10^{-4}$; Figs 2 and 4) when offer a choice to lay on plain purée. On the other hand, surprisingly, females showed a pronounced preference for purée with a lower MA concentration ($2.10^{-4}$ v/v), which kills half of the embryos (Fig 2). This narrow window of MA concentration which is both attractive to females for egg-laying and toxic to their eggs, raises the possibility that MA could be used for pest control.

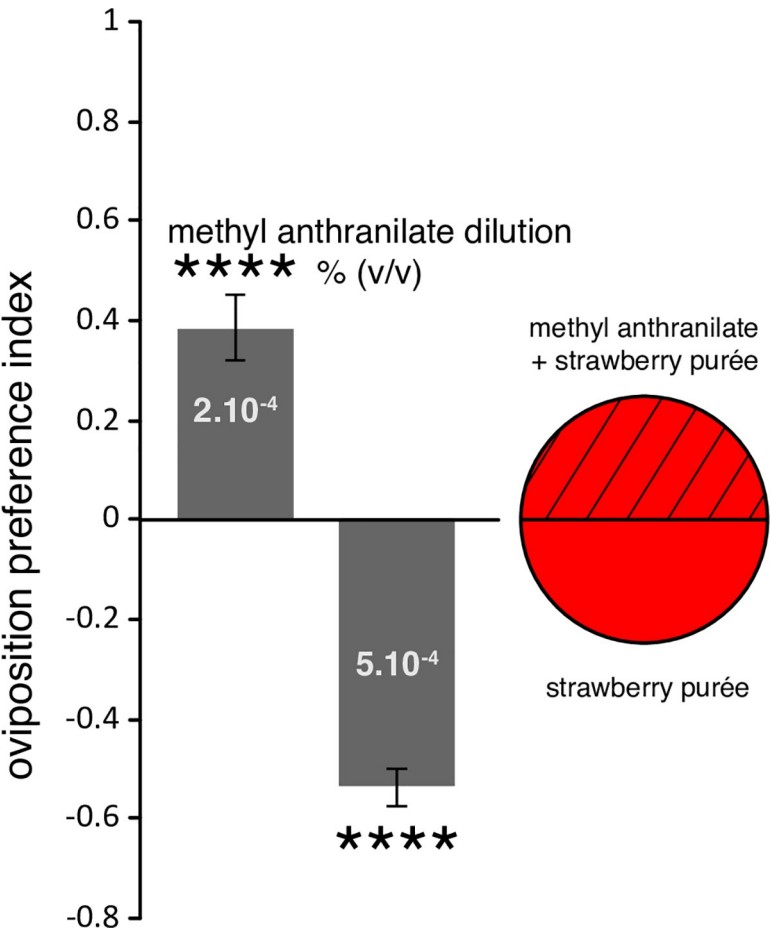

**Fig 4. *D. suzukii* egg-laying preference on MA substrate.** Female flies were offered the choice to lay eggs on strawberry purée supplemented with different concentration of MA or devoid of MA (egg-laying plate depicted to the right). MA at a concentration of $2.10^{-4}$ (v/v) is attractive for egg-laying, while it is repulsive at $5.10^{-4}$ (v/v). (Error bars represent SEM, n = 12 replicates per MA concentration, stars indicate significant difference from the hypothetical mean of 0; \*\*\*\*: P-value<0. 0005).

## The valence of volatile methyl anthranilate in *D. suzukii* is concentration-dependent

Positional preference and egg-laying preference do not always match in *Drosophila* [19, 20]. Given the potential interest of MA as a bioactive molecule and the female behavior toward it in the context of egg-laying, we further explored the attraction of adult *D. suzukii* for MA. Specifically, we wondered whether the egg-laying preference might be caused by a general positional preference associated to MA. In other words, flies would simply lay more eggs where they spend more time, and this could be conditioned by their attraction to or aversion for MA. To evaluate *D. suzukii* attraction to MA, we used a trap assay in which we offered flies the choice between entering a trap with water or a trap with water and MA (Fig 5). We found that the fly preference for MA depends on MA concentration. The odor presentation differs between the two assays and unlike in the egg-laying assay, flies perceive only plumes of volatile MA in the trap assay. The concentrations in the two assays are therefore not comparable. A concentration of $2.10^{-4}$ (v/v) results in a significant preference for MA over water alone (Fig 5). Higher concentrations, however trigger an aversive reaction (strong aversive trend at $4.10^{-4}$, P-value 0.0517). Although the context and MA concentrations cannot be strictly

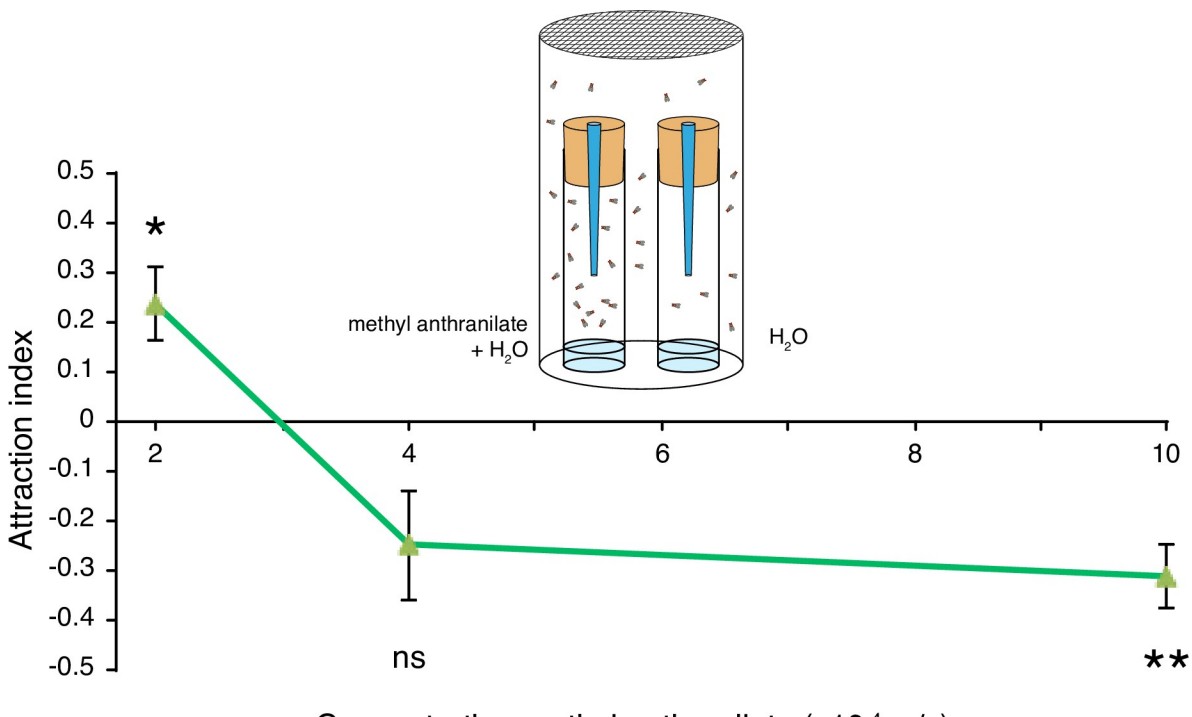

**Fig 5. Preference index for entering traps containing water or water supplemented with different concentrations of MA.** Flies displayed significant preference for traps containing $2.10^{-4}$ (v/v) MA over water. When testing a concentration of $1.10^{-3}$ (v/v) MA, flies preferred the water traps. No preference was detected when testing $4.10^{-4}$ (v/v) MA. (Error bars represent SEM, n = 9, stars indicate significant difference from the hypothetical mean of 0; *, P-value<0.05; **, P-value<0.005; ns: non-significant).

compared, these positional preferences are in line with the egg-laying preferences illustrated in Fig 4. It is therefore difficult to know whether MA concentration in a potential egg-laying substrate directly determines the choice of *D. suzukii* females, or whether they simply lay eggs where they are, or both. The former would likely involve their sense of taste, while the latter could rest exclusively on olfaction. We had shown, however, that *D. suzukii* females carefully chose their egg-laying substrate, using multiple sensory modalities [10]. We can at least conclude that females overall prefer to stay and lay their eggs in environments that do not jeopardize their offspring, but that a gray zone may exists where attractive MA is sufficiently concentrated to kill embryos.

## Discussion

In our previous study [7], we explored a large collection of strawberry accessions for their suitability to *Drosophila suzukii* larval development. We found that fruit size is one significant factor in determining how many adults emerge, with the accession identity being the other.

One of these accessions showed an unusual enrichment in MA and led to a significantly reduced emergence of flies. In the present study, we tried to elucidate whether MA in the fruits of these plants impacted fly survival. We showed that MA alone caused embryo lethality across a wide range of concentrations, and that this effect appears specific, as another comparable fruit compound, isoamyl acetate, was harmless.

Attempting to resolve the role of MA in fruits, we undertook a fractionation approach and met the limitations of this approach. Although we did find a correlation between the presence of MA and toxicity to the eggs, we faced the difficulty to establish a correspondence between

the real MA concentration in a fresh fruit, and the actual concentration in the fractions we tested. In spite of these difficulties, we think that fractionation, combined with toxicity tests, is one of the ultimate methods to track down which compounds carries the fruit bioactivity.

Our results also suggest a complex and paradoxical ecological relationship between *D. suzukii* and MA-enriched fruits. Females show a preference for substrates with MA at a concentration that is toxic and lethal for their offspring. Their preference to lay eggs on such toxic substrates could be explained by their general attraction to volatile MA at certain concentrations. This situation is reminiscent of *D. sechellia*'s attraction to lay eggs on *Morinda* fruits [21] whose high concentration of octanoic acid is toxic to the larvae of other species. The difference, though, is that *D. sechellia* larvae are surviving well on these fruits [21, 22]. While the pressure to utilize novel egg laying substrates is certainly higher in the island dwelling *D. sechellia* when compared with the now cosmopolitan *D. suzukii*, it does present another scenario of seemingly contradictory preferences in ovipositioning. The simplest hypothesis to resolve this paradox for *D. suzukii*, then, is that *D. suzukii* has not been exposed to MA-enriched fruits in its native geographic range. Therefore the 10–15 years of its worldwide spreading have not been sufficient to select *D. suzukii* that either dislike and avoid MA, or have become resistant to it.

This relationship between the invasive pest *D. suzukii* and MA might, however, represent an interesting opportunity for the fruit industry. Indeed, MA is not toxic to humans, and increasing its concentration on or in fruits that *D. suzukii* targets could be a pest management strategy. In principle, an increase of MA could be achieved by spraying (MA is relatively cheap), through crop selection, or through genetic engineering of specific crops, as the synthesis pathways for MA in fruits are at least partly known [16]. Alternatively, if fruits with increased MA concentration proved to be attractive to *D. suzukii* in the field, including for egg-laying, plants producing such fruits could be used as a sink to deplete local *D. suzukii* populations.

## Supporting information

**S1 Table. Sample weights, solvent amounts and extraction yields for fractionation.** (PDF)

## Author Contributions

**Conceptualization:** Lasse B. Bräcker, Xiaoyun Gong, Corinna Dawid, Klaus Olbricht, Martin Parniske, Nicolas Gompel.

**Data curation:** Lasse B. Bräcker, Xiaoyun Gong, Klaus Olbricht, Nicolas Gompel.

**Formal analysis:** Lasse B. Bräcker, Xiaoyun Gong, Klaus Olbricht, Martin Parniske, Nicolas Gompel.

**Funding acquisition:** Martin Parniske, Nicolas Gompel.

**Investigation:** Lasse B. Bräcker, Xiaoyun Gong, Christian Schmid, Corinna Dawid, Detlef Ulrich, Tuyen Phung, Alexandra Leonhard, Julia Ainsworth, Klaus Olbricht.

**Methodology:** Lasse B. Bräcker.

**Project administration:** Martin Parniske, Nicolas Gompel.

**Supervision:** Martin Parniske, Nicolas Gompel.

**Writing – original draft:** Lasse B. Bräcker, Nicolas Gompel.

                                      

**Writing – review & editing:** Xiaoyun Gong, Corinna Dawid, Detlef Ulrich, Klaus Olbricht, Martin Parniske.

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
