## [Decision Letter · Decision Letter 0]

15 Apr 2020

PONE-D-20-07692

A strawberry accession with elevated methyl anthranilate fruit concentration is naturally resistant to the pest fly Drosophila suzukii

PLOS ONE

Dear Dr. Gompel,

Thank you for submitting your manuscript to PLOS ONE. After careful consideration, we feel that it has merit but does not fully meet PLOS ONE’s publication criteria as it currently stands. Therefore, we invite you to submit a revised version of the manuscript that addresses the minor points raised during the review process.

We would appreciate receiving your revised manuscript by May 30 2020 11:59PM. To enhance the reproducibility of your results, we recommend that if applicable you deposit your laboratory protocols in protocols.io, where a protocol can be assigned its own identifier (DOI) such that it can be cited independently in the future. For instructions see: http://journals.plos.org/plosone/s/submission-guidelines#loc-laboratory-protocols

We look forward to receiving your revised manuscript.

Kind regards,

Frederic Marion-Poll, PhD

Academic Editor

PLOS ONE

Journal Requirements:

"The present work has been financed with funding from the

Ludwig-Maximilians University of Munich (NG and MP)."

"The authors received no specific funding for this work"

We note that one or more of the authors are employed by a commercial company: Hansabred GmbH & Co.

Please respond by return email with an updated Funding Statement and Competing Interests Statement and we will change the online submission form on your behalf.

4. We noted in your submission details that a portion of your manuscript may have been presented or published elsewhere.

"Figure 1A is adapted from one of our recent papers, as explicitely indicated in the text. It is important to show this panel in this context again, to compare it to the next 2 panels. The work is published in open access: " ext-link-type="uri" xlink:type="simple">https://www.frontiersin.org/articles/10.3389/fpls.2016.01880/full"

Reviewers' comments:

Reviewer's Responses to Questions

**Comments to the Author**

1. Is the manuscript technically sound, and do the data support the conclusions?

Reviewer #1: Yes

Reviewer #2: Yes

2. Has the statistical analysis been performed appropriately and rigorously? 

Reviewer #1: Yes

Reviewer #2: Yes

3. Have the authors made all data underlying the findings in their manuscript fully available?

Reviewer #1: Yes

Reviewer #2: Yes

4. Is the manuscript presented in an intelligible fashion and written in standard English?

Reviewer #1: Yes

Reviewer #2: Yes

5. Review Comments to the Author

Reviewer #1: Bräcker et al. A strawberry accession with elevated methyl anthranilate fruit concentration is naturally resistant to the pest fly Drosophila suzukii

PLOS ONE 3-28-20

The biological problem in this manuscript has agricultural and economical relevance. This problem is to understand the chemical basis of natural resistance to the crop pest D. suzukii. The authors find that one of the most resistant accessions of strawberries is abundant with methyl anthranilate. Methyl anthranilate alone or fractionated from the resistant strawberry accession decrease the survival of D. suzukii embryos. In contrast to its lethality on their embryos, methyl anthranilate attracts D. suzukii flies, which also prefer to lay eggs on substrates inoculated with this compound. This manuscript is well written and technically correct. Here are some minor comments:

1. The effect of this compound on adult survival requires investigation, especially all the behavior experiments were performed by adult flies. This isn’t necessarily for this study

2- In figure 2 caption isoamyl acetate is one word (isoamylacetate), please correct.

3. More concentrations are required to be tested for the oviposition experiments, try to be consistent with the concentrations used for the survival experiments.

4-. What is the amount of the purée in the egg-laying preference assay?

5- the same for the trap assay experiments, try to be consistent with the concentrations used for the survival experiments.

Reviewer #2: The authors found that a particularly resistant strawberry accession to Drosophila suzukii had a high concentration of methyl anthranilate (MA). Through several accurate bioassays including different concentrations, the authors demonstrated that MA is indeed toxic to the embryos, in spite of its olfactory attraction to the females and their preference to lay eggs on media with low toxic concentrations. This result is very interesting and very promising for agriculturalists as well as scientists interested in the biology of insect-plant interactions.

While I recommend the acceptance of the paper as it is, I believe that two additional points may be raised or discussed by the authors. The first point considers the relative significance of MA in strawberry resistance. The authors noted that the MA-based resistance is unique to a single accession indicating that variable resistance mechanisms may exist in different strawberry cultivars and species. In a previous paper, they showed that emergence was correlated with fruit size. It might hence have been preferable to show on a graph whether the resistant accession P300 has also a smaller size (e.g., a 3D plot with MA concentration, fruit size and average emergence per strain/species). The authors had also measured the concentration of other substrates for each strain/species. A multivariate regression analysis of average emergence over the different chemical concentrations, water loss (if measured for all strains/species) and fruit size could also be helpful to explain how much MA could explain fruit resistance.

The second point considers the significance for MA adaptation in D. suzukii. The authors compared this situation (a female preference to a toxic substance) with D. sechellia adaptation to Morinda citrifolia. However, D. sechellia is an island species and it might have been obliged to adapt to its host. For D. suzukii, the geographical distribution and the niches are far broader. High MA concentration may not be as frequently encountered to necessitate particular genetic adaptation, either in native or introduced populations.

6. PLOS authors have the option to publish the peer review history of their article (what does this mean?). If published, this will include your full peer review and any attached files.

Reviewer #1: Yes: Hany K. M. Dweck

Reviewer #2: No

---

## [Author Response · Author response to Decision Letter 0]

12 May 2020

A point by point response to the reviewers is enclosed in the revised submission.

---

## [Editor Report · Decision Letter 1]

19 May 2020

A strawberry accession with elevated methyl anthranilate fruit concentration is naturally resistant to the pest fly Drosophila suzukii

PONE-D-20-07692R1

Dear Dr. Gompel,

We are pleased to inform you that your manuscript has been judged scientifically suitable for publication and will be formally accepted for publication once it complies with all outstanding technical requirements.

With kind regards,

Frederic Marion-Poll, PhD

Academic Editor

PLOS ONE

Additional Editor Comments (optional):

The authors have answered all questions of the reviewers.
---

## [Editor Report · Acceptance letter]

22 May 2020

PONE-D-20-07692R1 

A strawberry accession with elevated methyl anthranilate fruit concentration is naturally resistant to the pest fly *Drosophila suzukii*

Dear Dr. Gompel:

I am pleased to inform you that your manuscript has been deemed suitable for publication in PLOS ONE. Congratulations! Your manuscript is now with our production department. 

With kind regards,

on behalf of

Professor Frederic Marion-Poll 

Academic Editor

PLOS ONE